# Different contra-sound effects between noise and music stimuli seen in N1m and psychophysical responses

Masayuki Shirakura[1], Tetsuaki Kawase[1,2,3]*, Akitake Kanno[4,5], Jun Ohta[1], Nobukazu Nakasato[4,5], Ryuta Kawashima[6], Yukio Katori[1]

**1** Department of Otolaryngology-Head and Neck Surgery, Tohoku University Graduate School of Medicine, Sendai, Miyagi, Japan, **2** Laboratory of Rehabilitative Auditory Science, Tohoku University Graduate School of Biomedical Engineering, Sendai, Miyagi, Japan, **3** Department of Audiology, Tohoku University Graduate School of Medicine, Sendai, Miyagi, Japan, **4** Department of Electromagnetic Neurophysiology, Tohoku University School of Medicine, Sendai, Miyagi, Japan, **5** Department of Epileptology, Tohoku University Graduate School of Medicine, Sendai, Miyagi, Japan, **6** Institute of Development, Aging and Cancer, Tohoku University, Sendai, Miyagi, Japan

* kawase@orl.med.tohoku.ac.jp

**Data Availability Statement:** All relevant data are within the paper and its Supporting Information files.

## Abstract

Auditory-evoked responses can be affected by the sound presented to the contralateral ear. The different contra-sound effects between noise and music stimuli on N1m responses of auditory-evoked fields and those on psychophysical response were examined in 12 and 15 subjects, respectively. In the magnetoencephalographic study, the stimulus to elicit the N1m response was a tone burst of 500 ms duration at a frequency of 250 Hz, presented at a level of 70 dB, and white noise filtered with high-pass filter at 2000 Hz and music stimuli filtered with high-pass filter at 2000 Hz were used as contralateral noise. The contralateral stimuli (noise or music) were presented in 10 dB steps from 80 dB to 30 dB. Subjects were instructed to focus their attention to the left ear and to press the response button each time they heard burst stimuli presented to the left ear. In the psychophysical study, the effects of contralateral sound presentation on the response time for detection of the probe sound of a 250 Hz tone burst presented at a level of 70 dB were examined for the same contra-noise and contra-music used in the magnetoencephalographic study. The amplitude reduction and latency delay of N1m caused by contra-music stimuli were significantly larger than those by contra-noise stimuli in bilateral hemisphere, even for low level of contra-music near the psychophysical threshold. Moreover, this larger suppressive effect induced by contra-music effects was also observed psychophysically; i.e., the change in response time for detection of the probe sound was significantly longer by adding contralateral music stimuli than by adding contra-noise stimuli. Regarding differences in effect between contra-music and contra-noise, differences in the degree of saliency may be responsible for their different abilities to disturb auditory attention to the probe sound, but further investigation is required to confirm this hypothesis.

**Funding:** a) Grants received by: TK b) Grant numbers: Grant-in-Aid for Exploratory Research, No. 18K19597; and Grant-in-Aid for Scientific Research (B), No. 20H03831. c) Name of Funder: The Ministry of Education, Culture, Sports, Science and Technology, Japan d) URL of the Funder: https://www.mext.go.jp/a_menu/shinkou/hojyo/main5_a5.htm e) The funder had no role in the study design, data collection and analysis, decision to publish, or preparation of the manuscript.

**Competing interests:** The authors have declared that no competing interests exist.

## Introduction

In the phenomenon of auditory masking, the audibility of a signal is decreased by the presence of another sound (masker); i.e., the detection threshold for the signal is elevated and the auditory-evoked responses to the signal sound are reduced by presentation of the masker [1–6]. Any level of the auditory pathway from the cochlea to the cortex can be involved in this masking phenomenon [2,3,5,7]. Thus, even when the masker presents to the contralateral ear, the ipsilateral response could be affected via a central masking mechanism [2–4].

One possible method for investigating central masking is to examine the contralateral masking phenomenon observed in auditory cortical-evoked responses such as the N1m response, which is an evoked wave occurring with a post-stimulus latency of approximately 100 ms, and generated mainly from the primary and secondary auditory cortices located in the superior temporal gyrus in Heschl's gyrus and the planum temporale [8–16]. In other words, sound presented to the contralateral ear can affect the auditory N1m in response to a signal tone presented to the ipsilateral ear via a central masking mechanism. However, the strength of the contralateral masking effect appears to differ according to the characteristics of the contra-masking sound; i.e., the N1m response is not significantly affected by contralateral continuous white noise [4,17,18], but is significantly suppressed by speech sound, music sound, and intermittent noise [4]. These results hint at the possible involvement of a factor other than a simple "masking" phenomenon in suppression of N1m caused by such as music, speech and intermittent noise, but the underlying mechanism for the different contra-effects of these stimuli remains unclear [4]. In addition, this previous study of Hari and Mäkelä [4] indicated the effect of contralateral sound on the N1m amplitude obtained from the right hemisphere for one particular sound pressure level of contra-lateral sounds, but more detailed features of these contra-sound effects such as those on N1m latencies, the effects of level of contra-sound on the magnitude of contra-sound effects (i.e., whether or not the N1 suppression effect caused by the contralateral sound is a phenomenon that depends on the presentation level of the contralateral sound), and inter-hemispheric differences have not yet been fully clarified [4].

Moreover, auditory-evoked responses can be affected by the sound presented to the contralateral ear, through peripheral mechanisms that include masking due to cross-talk and the olivocochlear (OC) efferent system that innervates the cochlear and/or the middle ear muscle system, in addition to the central masking mechanism that occurs in the brain [19–27]. Thus, it is necessary to minimize these peripheral effects during observation of the central masking effects caused by contra-sound effects, but it appears that previous studies have given little consideration to this requirement.

Against this background, the focus of the present study was to clarify in detail the features of the relatively larger contra-music effects than the contra-noise effects studied previously [4], which presumably occur mainly in the brain, while minimizing and/or assessing the possible peripheral effects caused by the presentation of contra-sound. Taking into consideration the level and/or frequency characteristics of the stimuli used in order to minimize the cross-talk effects as well as OC and MEMs effects, the effects of contra-music were compared with those of contra-noise on the latency and amplitude of N1m for various levels of contra-sound using magnetoencephalography (MEG), which can separate the activation of auditory cortices in the right and left hemispheres [8–16].

Furthermore, to examine whether the phenomena observed in the N1 response are also observed psychophysically, the different contra-effects between noise and music were also examined psychophysically, based on the results of the MEG study.

## Materials and methods

All experiments were approved by the ethical committee of the Tohoku University Graduate School of Medicine (#2020-1-597 [MEG study] and #2020-1-641 [psychophysical study]) and written informed consent was obtained from each subject in accordance with the requirements of the ethical committee. All aspects of the study were performed in accordance with the guidelines of the Declaration of Helsinki (1991).

### MEG study

**Subjects.**  The subjects in the MEG study were 12 normal volunteers (12 males; mean age ± standard deviation (SD), 37.3 ± 11.2 years) with no history of auditory disease or neurological disorder. All were right-handed with Edinburgh Handedness Inventory score > +90 [28].

**Stimuli.**  The stimulus to elicit the N1m response was a tone burst of 500 ms duration (rise and fall time, 10 ms; plateau time, 480 ms) at a frequency of 250 Hz, presented to the left ear at a level of 70 dB SPL with an inter-stimulus interval of 3 s (0.33 Hz).

Different to the study of Hari and Mäkelä [4], filtered white noise (white noise filtered with a high-pass filter (filter slope = 24 dB/octave) at 2000 Hz) and filtered music stimuli (music stimuli filtered with a high-pass filter (filter slope = 18 dB/octave) at 2000 Hz) were presented to the right ear as contralateral noise. The frequencies of the probe tone (250 Hz) and the cut-off frequency of contra-sound were separated by 3 octaves to minimize the direct masking effects by the contra-sound on the probe tone, taking the following into consideration: the possible maximum cross-talk level of the 250 Hz component expected from the level of maximum sound pressure level of the filtered contra-stimuli, the filter slope the used in the present study, and the possible inter-aural attenuation level [29]. The music stimuli was a jazz piano piece ("You Took Advantage of Me") by Art Tatum, as used in previous studies because of its abundant intensities and frequency transitions, and because its large contra-sound effects have been confirmed previously [4].

**Procedure.**  Fig 1 shows a schema of the experimental protocol. Basically, N1m responses to the tone burst with and without contralateral continuous stimuli were recorded alternately while decreasing the level of contra-stimuli from 80 dB SPL to 30 dB SPL in 10 dB steps, and the relationship between the level of the contra-stimuli and the magnitude of N1m suppression was examined. In all cases, the effects of contra-music on the N1m responses were measured first, and then the effects of contra-noise were measured, but in most cases, the contra-music and contra-noise experiments were performed on different days, considering the mental and physical burden that they placed on the subjects. Measurements to assess the effects of each of contra-music and contra-noise were obtained continuously from 80 dB to 30 dB as much as possible. However, in case that the measurements needed to be paused for some reason, they were suspended after measurement in the control condition (i.e., N1m measurements without contralateral sound) and restarted from the same condition without contralateral sound. Subjects were instructed to focus their attention on the left ear and to press the response button whenever they heard burst stimuli presented to the left ear. These responses were recorded together with continuous MEG recordings.

**Acoustic reflex and psychophysical threshold for contra-stimuli.**  After the MEG measurements, acoustic reflex (sound-evoked middle ear reflex) and psychophysical threshold to the contralateral stimuli (noise and music) were then assessed. The contralateral acoustic reflex in response to the filtered noise and music presented to the right ear was measured using a commercially available impedance audiometer (RS-33; RION, Kokubunji, Tokyo, Japan) from 80 dB SPL in 10 dB steps, and the acoustic reflex threshold (ART) (visual detection threshold)

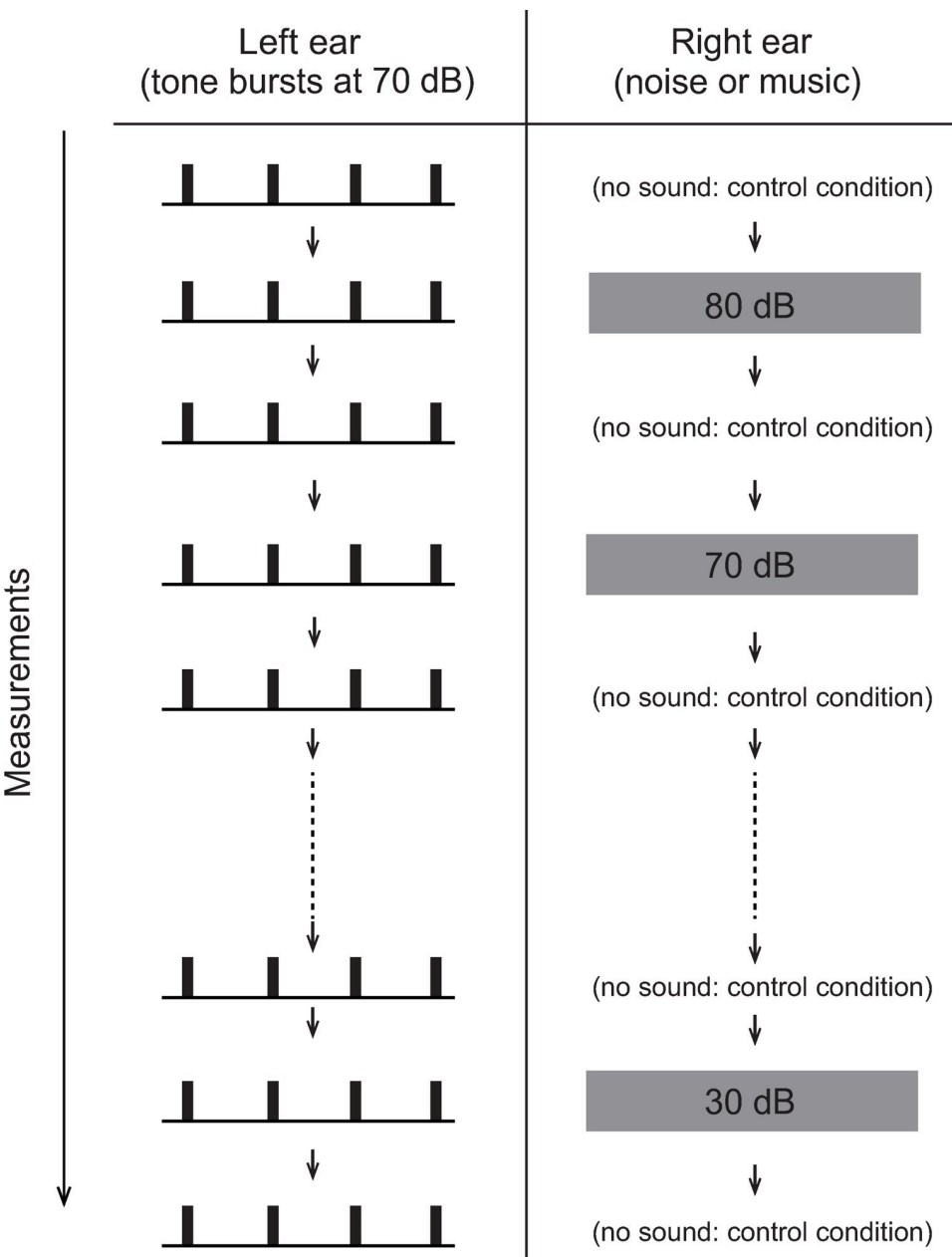

**Fig 1. Schema of the experimental protocol.** The N1m responses to tone bursts presented to the left ear at a level of 70 dB with and without contralateral stimuli presented continuously were recorded alternately while decreasing the level of contra-stimuli from 80 to 30 dB. In most cases, the contra-music and contra-noise experiments were performed on different days, considering the mental and physical burden that they placed on the subjects.

was determined as the minimum sound level to obtain the sound-evoked response. The psychophysical thresholds for each of filtered music and noise were measured in 5 dB steps and determined as the minimum sound pressure level to perceive the sound.

**Recording and analysis.** Auditory evoked fields (AEFs) were recorded in a magnetically shielded room using a 200-channel whole-head type axial gradiometer system (MEGvision PQA160C-RO; Ricoh, Tokyo, Japan). The sensors are configured as first-order axial gradiometers with a baseline of 50 mm; each gradiometer coil is 15.5 mm in diameter. The sensors are

arranged in a uniform array on a helmet-shaped surface at the bottom of a dewar vessel, and the mean distance between the centers of two adjacent coils is 25 mm. Sensor field sensitivity (noise of the system) was 3 fT/Hz within the frequency range.

AEFs were recorded only in the wake state as confirmed by real-time monitoring of the occipital alpha rhythm by MEG. The MEG signal was band-pass filtered between 0.03 Hz and 400 Hz (filter slope = 12 dB/octave) sampled at 10,000 Hz.

All MEG signals were continuously recorded during the entire experimental duration, and later analyzed (offline) using the built-in software in the MEG system (MEG Laboratory, Ricoh). To obtain the N1m response to tone bursts, the data from 50 ms before to 450 ms after the stimulus onset were averaged 50–100 times. In subsequent off-line analysis, the averaged data were digitally band-pass filtered from 2.0 to 45.0 Hz (setting conditions for the high-pass filter: cut-off frequency = 2.0 Hz, filter type = hamming window, filter width = $1.29 \times 2$ pi; setting conditions for the low-pass filter: cut-off frequency = 45.0 Hz, filter type = hamming window, filter width = $29 \times 2$ pi). The N100m response was identified visually as the first prominent negative peak at 80–120 ms after the onset, with the iso-field map showing downward current orientation. The location of each source was estimated at the N100m peak latency, using an equivalent current dipole (ECD) model with the best fit sphere for each subject's head. The source was superimposed on a three-dimensional MR image of the individual subject using a MEG–MR image coordination integration system.

The contra-sound effects on N1m latency and amplitude were assessed based on the averaged wave (root mean square [RMS] wave) of all channels in each hemisphere by comparing those obtained under each level of contralateral sound with the average values of those obtained from N1m without contralateral sound, which were measured just before and after each measurement condition with contralateral sound.

**Statistics.** Differences between the contra-music and contra-noise effects on N1m amplitudes and latencies were determined by two-way analysis of variance (ANOVA) with Bonferroni post-hoc analysis for multiple comparisons using SPSS software (ver. 26; IBM, Armonk, NY, USA). Values of $p < 0.05$ were considered to be significant.

## Psychophysical measurements on contra-sound effects

**Subjects.** Psychophysical measurements were performed in 15 normal volunteers (11 males, 4 females; mean age ± SD, 36.3 ± 10.8 years) with no history of auditory disease or neurological disorder. Nine of these subjects also participated in the MEG study.

**Methods.** The influence of contralateral sound presentation on the reaction time for detection of probe sound of a 250 Hz tone burst (duration, 500 ms; rise–fall time, 1 ms) presented at a level of 70 dB SPL was examined. Subjects were instructed to press the response button as soon as they detected each of the probe sounds, which were presented serially 60 times (one session) at a rate of approximately one every 1500 ms. The response button was pressed with whichever finger the volunteer found easiest, with either the right or left hand. The timing of presentation of the 250 Hz probe tones was formatted as a wav file and controlled by a PC system. The captured reactions were also recorded using the PC system. Because the first few trials can be unstable, depending on the subject, the first five trials were discarded and the average reaction time for the subsequent 55 trials was recorded as the reaction time for each measurement condition. As in the MEG study, the probe tone was presented to the left ear and the contra-sound (2000 Hz high-pass filtered white noise or 2000 Hz high-pass filtered music stimuli) was presented continuously to the right ear during each of 60 measurements (one session) through a headphone system (MDR-CD900ST; Sony, Tokyo, Japan)

via USB interfaces (Rubix 22; Roland, Hamamatsu, Japan; and Komplete Audio 6; Native Instruments, Berlin, Germany).

After a practice session, reaction times for detecting the probe tone were measured under the following three conditions: 1) without contralateral sound, 2) with contralateral 2000 Hz high-pass filtered music presented continuously at 60 dB, and 3) with 2000 Hz high-pass filtered white noise presented continuously at 60 dB. The order of these three protocols was adjusted to counterbalance the order effect among the subjects.

**Statistics.**  Differences between the contra-music and contra-noise effects on the psychophysical response were assessed by paired t-test using SPSS software (ver. 26; IBM). Values of p < 0.05 were considered to be significant.

## Results

### MEG study

Figs 2 and 3 show representative examples of the effects of contralateral "music" and "noise" stimuli at 50 dB, respectively, on the N1m response obtained from the right hemisphere. Superimposed magnetic signals with and without contra-sound stimuli (Figs 2A and 3A) are shown with isofield maps and ECDs superimposed on MR images (Figs 2B and 3B, without contra-sound stimuli; and Figs 2C and 3C, with contra-sound stimuli). When the averaged waves (root mean square [RMS] waves) of all channels in the right hemisphere measured with contra-music stimuli were superimposed with those of the control measurements (without contra-sound) obtained just before and just after those measurements (Fig 2D), the amplitude and latency of the N1m response were clearly reduced and delayed, respectively. In contrast, little effect was observed when noise stimuli were used as the contra-sound (Fig 3D). In the following analysis, the effects of contra-sound on the latency and amplitude of N100m of the averaged wave (RMS waves) of all channels in each hemisphere were compared between the effects of contra-music and contra-noise. The contra-sound effects on the N1m latency and amplitude (change in amplitude and in latency) were assessed by comparing the amplitude and the latency of N1m between those obtained under each level of contralateral sound and the average of those obtained from the two control measurements just before and after each measurement condition with contralateral sound.

Fig 4 shows the comparison of the effects of contra-sound on N1m amplitudes (Fig 4A) and latencies (Fig 4B) obtained from the right hemisphere between music and noise stimuli for all 12 subjects. Two-way ANOVA with Bonferroni post-hoc analysis was performed with contra-sound (noise and music) and the contra-level (30, 40, 50, 60, 70, and 80 dB) as the main factors. The results of the two-way ANOVA for the effects of contra-sound on N1m amplitude revealed insignificant sound-by-level interaction ($F(5, 132) = 0.720$, $p = 0.610$, $\eta^2 = 0.186$); a significant main effect for contra-sound ($F(1,132) = 48.354$, $p < 0.001$, $\eta^2 = 0.239$), indicating that N1m amplitude was significantly decreased by the addition of contra-music compared with contra-noise effects; and a significant main effect for contra-sound level ($F(5,132) = 3.767$, $p = 0.003$, $\eta^2 = 0.093$), indicating that the contra-sound effects on N1m amplitude showed a tendency to increase with increasing contra-sound level. Bonferroni post-hoc analysis revealed that contra-music effects on N1m amplitude reduction were significantly larger than those of contra-noise, for all contra-sound levels ($p < 0.05$) (Fig 4A). Differences of amplitude reduction induced by the contra-sounds between levels were significant between contra-music effects obtained for 30 dB and those for 70 as well as 80 dB.

In contrast, two-way ANOVA with Bonferroni post-hoc analysis for effects on latency, performed with contra-sound (noise or music) and contra-level (30, 40, 50, 60, 70, and 80 dB) as the main factors, revealed an insignificant sound-by-level interaction ($F(5, 132) = 0.317$,

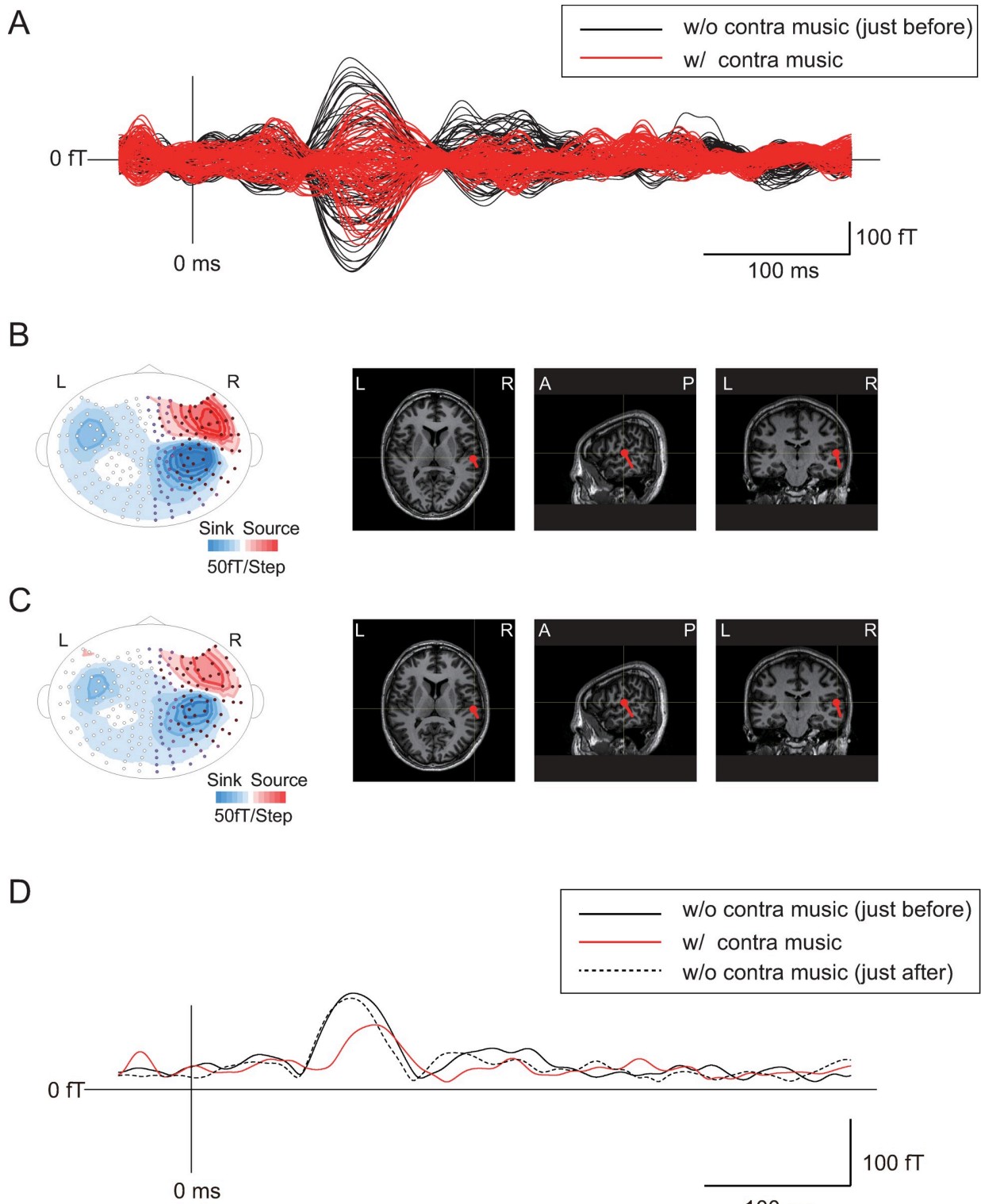

**Fig 2. Representative example of the effects of contralateral "music" stimuli (at 50 dB) on the N1m response obtained from the right hemisphere.** A: Superimposed magnetic signals with and without contra-music stimuli, B: Isofield maps and ECDs superimposed on MR images without contra-music stimuli, C: Isofield maps and ECDs superimposed on MR images with contra-music stimuli, D: Averaged waves (root mean square [RMS] waves) of all channels in the right hemisphere measured with contralateral sound were superimposed with those of the control measurement (measurement without contra-sound) just before and after the measurement with contralateral sound.

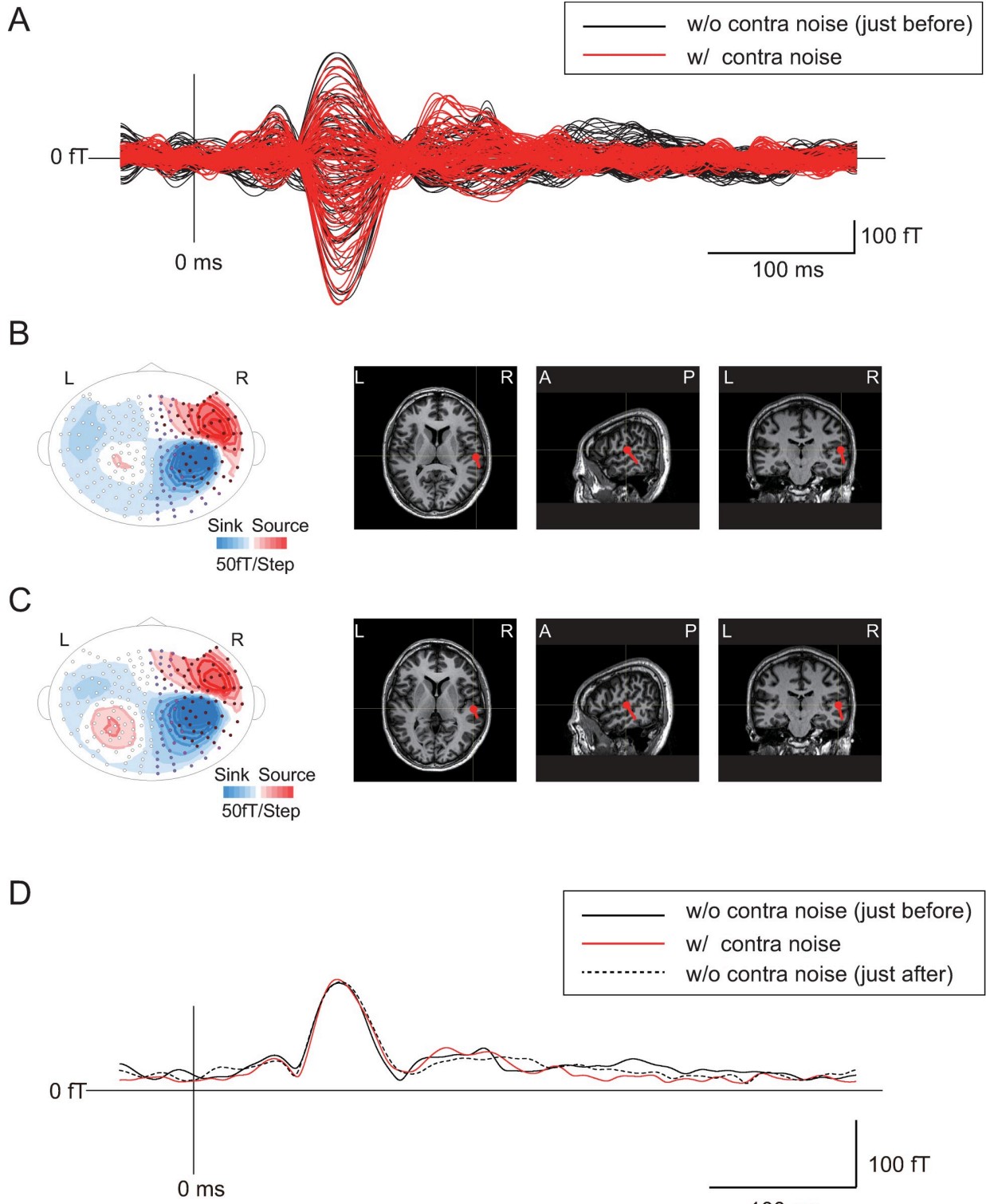

**Fig 3. Representative example of the effects of contralateral "noise" stimuli (at 50 dB) on the N1m response obtained from the right hemisphere.** A: Superimposed magnetic signals with and without contra-noise stimuli, B: Isofield maps and ECDs superimposed on MR images without contra-noise stimuli, C: Isofield maps and ECDs superimposed on MR images with contra-noise stimuli, D: Averaged waves (root mean square [RMS] waves) of all channels in the right hemisphere measured with contralateral sound were superimposed with those of the control measurement (measurement without contra-sound) just before and after the measurement with contralateral sound.

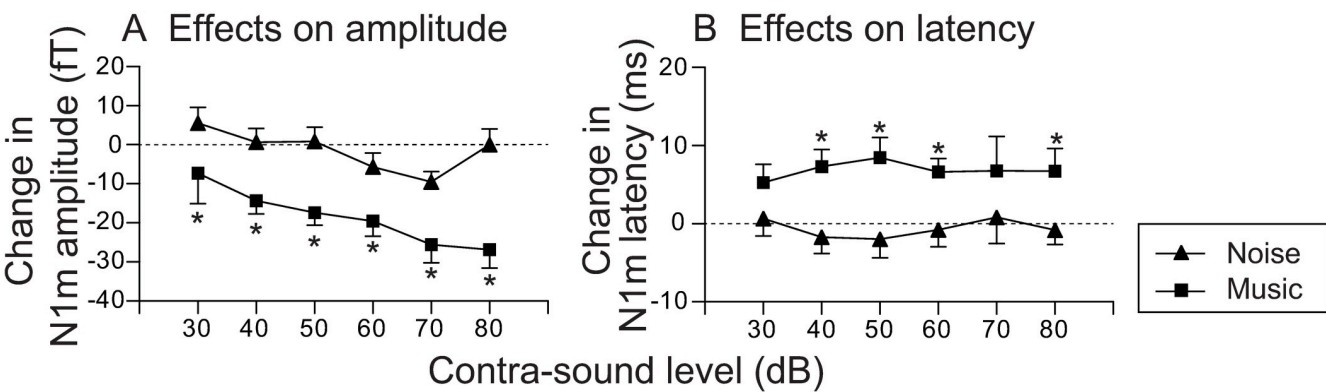

**Fig 4.** Average effect of contra-sound on N1m amplitude (A) and latency (B) obtained from the right hemisphere for each contra-sound level. Values are compared between music and noise stimuli for all 12 subjects. Error bars indicate standard error (SE). Asterisks indicate significant difference (p<0.05) between the effects of contra-music and contra-noise for each contra-sound level (Bonferroni post-hoc analysis of ANOVA) (see text for further details on statistics).

$p = 0.902$, $\eta^2 = 0.010$); a significant main effect for contra-sound (F(1,132) = 24.619, $p < 0.001$, $\eta^2 = 0.155$), indicating that N1m latencies were significantly delayed by the addition of contra-music compared to contra-noise effects on N1m latencies; and an insignificant main effect for contra-sound level (F(5,132) = 0.039, $p = 0.999$, $\eta^2 = 0.001$), indicating that contra-sound effects on N1m latencies were not significantly different among the different contra-sound levels. Bonferroni post-hoc analysis revealed that contra-music effects on the delay of N1m latency were significantly larger than those of contra-noise for contra-sound levels of 40, 50, 60, and 80 dB ($p < 0.05$) (Fig 4B).

Fig 5 shows the comparison of the effects of contra-sound on N1m amplitudes and latencies obtained from the left hemisphere between music and noise stimuli. As some of the N1m waves obtained from the left auditory cortex were too small to analyze, in the present study

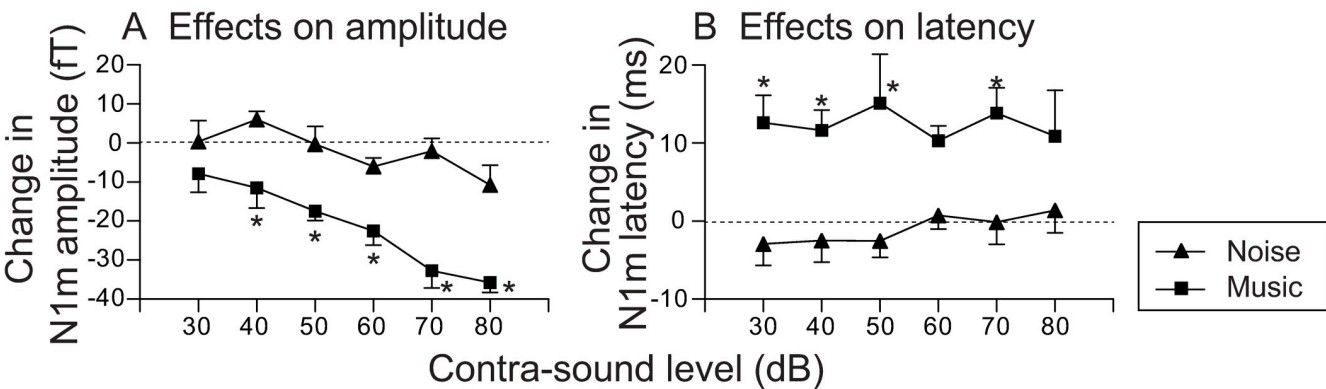

**Fig 5.** Average effect of contra-sound on N1m amplitude (A) and latency (B) obtained from the left hemisphere for each contra-sound level. Values are compared between music and noise stimuli for 8 subjects in whom the latency and amplitude of N1m could be assessed reliably under all measurement conditions. Error bars indicate standard error (SE). Asterisks indicate significant difference (p<0.05) between the effects of contra-music and contra-noise in each contra-sound level (Bonferroni post-hoc analysis) of ANOVA (see text for further details on statistics).

only those in which assessment was performed reliably under all measurement conditions (data of 8 subjects) were included for analysis. Two-way ANOVA with Bonferroni post-hoc analysis for effects on amplitude was also performed with contra-sound (noise or music) and contra-level (30, 40, 50, 60, 70, and 80 dB) as the main factors. The results of two-way ANOVA indicated insignificant sound-by-level interaction (F(5, 84) = 1.897, $p$ = 0.103, $\eta^2$ = 0.046); a significant main effect for contra-sound (F(1,84) = 69.949, $p$ < 0.001, $\eta^2$ = 0.342), indicating that N1m amplitude was significantly decreased by the addition of contra-music compared to the contra-noise effects on N1m amplitude; and a significant main effect for contra-sound level (F(5,84) = 8.228, $p$ < 0.001, $\eta^2$ = 0.201), indicating that contra-sound effects on N1m amplitude tended to increase as contra-sound level increased. Bonferroni post-hoc analysis revealed that contra-music effects on N1m amplitude reduction were significantly larger than those by contra-noise for all contra-sound levels except for 30 dB ($p$ < 0.05) (Fig 5A). Differences of amplitude reduction induced by contra-sounds between levels were significant only between contra-music effects obtained for 30 dB and those for 70 and 80 dB, and between those for 40 dB and those for 80 dB.

In contrast, two-way ANOVA with Bonferroni post-hoc analysis for effects on latency, which was performed with contra-sound (noise and music) and contra-level (30, 40, 50, 60, 70, and 80 dB) as the main factors, revealed insignificant sound-by-level interaction (F(5, 84) = 0.436, $p$ = 0.822, $\eta^2$ = 0.017); a significant main effect for contra-sound (F(1,84) = 44.033, $p$ < 0.001, $\eta^2$ = 0.337), indicating that N1m latencies were significantly delayed by the addition of contra-music compared to contra-noise effects on N1m latencies; and an insignificant main effect for contra-sound level (F(5,84) = 0.128, $p$ = 0.986, $\eta^2$ = 0.005), indicating that contra-sound effects on N1m latencies were not significantly different among the different contra-sound levels. Bonferroni post-hoc analysis revealed that contra-music effects on the delay of N1m latency were significantly larger than those of contra-noise for contra-sound levels of 30, 40, 50, and 70 dB ($p$ < 0.05) (Fig 5B).

Fig 6 shows the ART and psychophysical thresholds for contra-music and contra-noise. Significant amplitude reduction and latency delay of N1ms were observed for contra-music, at levels below those of ART and near the psychoacoustic threshold.

## Psychophysical study

The MEG study revealed that the amplitude and latency of N1m in response to 250 Hz tone burst at 70 dB SPL were reduced and delayed, respectively, by the presentation of music material to the contralateral ear. It was speculated that these observations in the N1m response may affect the sensation of loudness and detection latency for the 70 dB tone burst in a psychophysical manner. However, as it appeared that it would be difficult to accurately evaluate slight changes in loudness sensation due to the presence or absence of consonant sound, we decided to first observe the effect on latency. The influence of contralateral sound presentation on the response time to the probe sound presented to the left ear was examined. The contra-music effect on the response time to the probe tone (relative response time to those measured without contra-sound) was significantly larger than the contra-noise effect (Fig 7).

## Discussion

The major findings of the present study are as follows: 1) contra-music stimuli caused significantly larger amplitude reduction and latency delay of N1m compared with those by contra-noise stimuli; 2) these contra-music effects could be observed even below the ART, and even near the level of psychoacoustic thresholds for the contra-sound stimuli, basically in the bilateral hemispheres; 3) the effects of contra-music level on each of N1m amplitude and latency

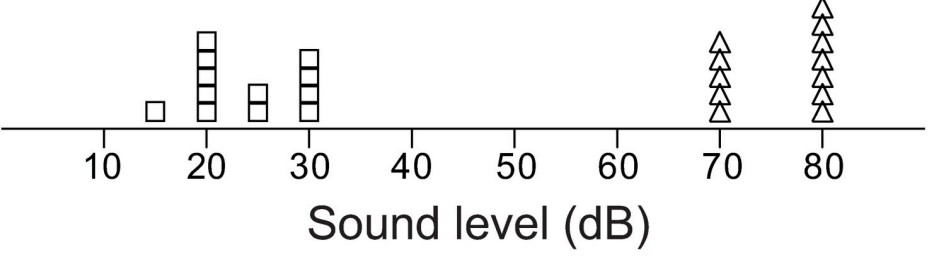

## A  For music stimuli

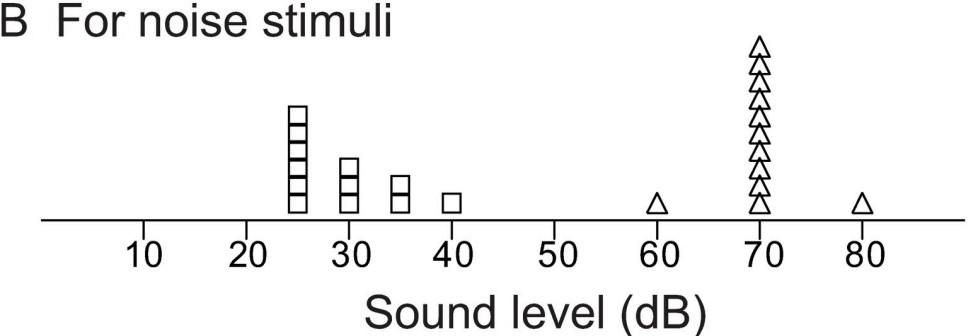

## B  For noise stimuli

**Fig 6. Acoustic reflex thresholds and psychophysical thresholds for contra-music and contra-noise stimuli for participants in the MEG study.** The upper figure (A) shows the acoustic reflex threshold (open triangles) and psychoacoustic threshold (open squares) for music stimuli, and the lower figure (B) shows the acoustic reflex threshold (open triangles) and psychoacoustic threshold (open squares) for noise stimuli.

differed in the following regards: those on N1m latency were relatively constant regardless of the level of contra-music, whereas those on N1m amplitude showed a greater increase as the level of contra-music increased; 4) these larger contra-music effects compared with contra-noise effects were also observed psychophysically; i.e., response time to the probe tone was significantly delayed by the presentation of contra-music stimuli.

Auditory-evoked responses can be affected by the sound presented to the contralateral ear, through peripheral mechanisms such as the olivocochlear (OC) efferent innervation of the cochlear and/or middle ear muscle system [29] and also by a central mechanism. However, it is thought that most of the effects of contralateral music stimuli on the N1m revealed in the present study were mediated via the central nervous system rather than a peripheral mechanism, for the following reasons.

Contralateral sound can cause considerable stimulation of the OC efferent system [19–23]. In general, OC-mediated effects are remarkable in relatively higher frequency regions (higher than 2000–4000 Hz) in animals [29]; however, it is known that OC-mediated effects can be observed in the low frequency area in addition to the higher frequency area in humans [30–

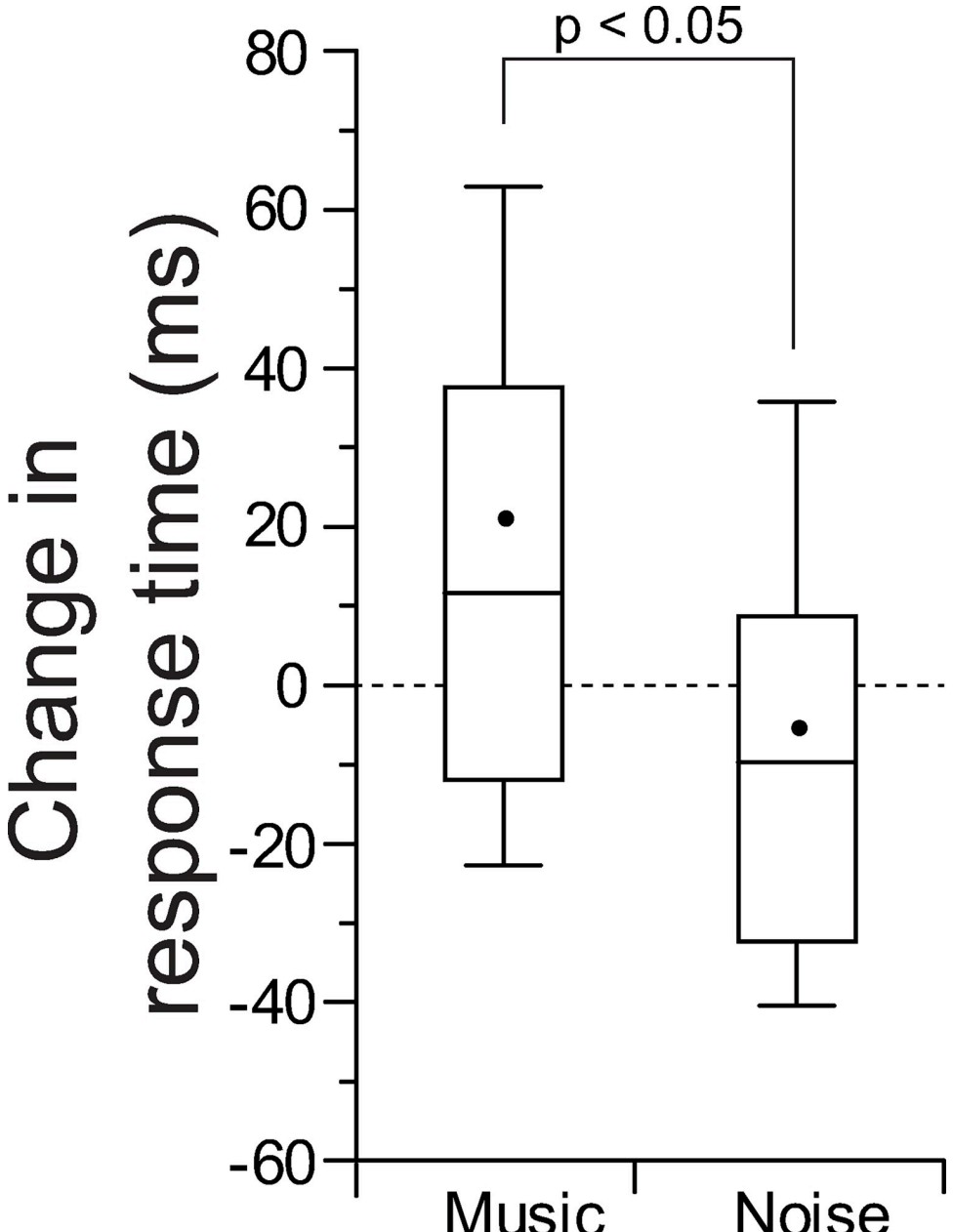

**Fig 7. Effects of the contralateral music stimuli vs. noise stimuli on the response time to the probe sound presented to the left ear ("change in response time" indicates the response time relative to those measured without contra-sound).** The black dots in the box-plots indicate the mean values.

35]. In this respect, response to the 250 Hz probe tone in humans may be affected by the contra-sound via the OC-efferent system. However, considering that OC-effects are well elicited by broad-band noise including the probe-tone frequency [22,23,35], we speculate that it is less possible that the 2 kHz high-pass filtered contra sound used in the present study could have evoked the considerable OC-mediated effects on the probe tone at 250 Hz. In fact, no considerable effects were found in N1m responses against the 2 kHz filtered contra-noise in the present study. Based on these known characteristics of OC-mediated effects, the contribution of

the OC efferent system to the suppressive effect on N1m that was observed due to contra-music in the present study is thought to be minimal. On the other hand, the frequency characteristics of middle ear muscle contraction are known to be remarkable in the lower frequency region around 125–500 Hz [24–27,30]. The ART in response to contralateral music and noise stimuli in the present study suggests the possible involvement of ART in contra-music effects by a high-level elicitor at 80 or 70 dB; however, as shown in Fig 6, few of the contra-music effects below 60 dB would be associated with ART. The small contra-effects observed for high-level filtered noise, which could be elicited in part by ART, also suggest little contribution of the middle ear muscles to the contra-music effects elicited by high-level contra-music.

Another possible suppressive peripheral mechanism is a masking phenomenon due to cross-talk. Although the high-level contra-sound used in the present study (70–80 dB) could cause cross-talk effects, the masking effects of the 2000-Hz high-pass sound on the 250 Hz tone burst would be negligible when taking into consideration the upward masking pattern in the periphery (as the masking effects basically spread toward regions of higher frequency than the masker frequency), as well as the frequency separation between the 250 Hz tone-burst to elicit N1m and the 2000-Hz high-pass filtered contra-sound used in the present study. Therefore, it appears less likely that a peripheral mechanism is involved in the contra-music effects observed in the present study.

In contrast, a psychophysically observed classical central masking effect is well known as a possible mediator of contra-sound effects as a central mechanism [3] but may also be reduced due to the frequency characteristics of this phenomenon and the frequency separation between the tone burst and the contra-music. In addition, if this mechanism is involved, contra-effects should be also observed for contra-noise, and the contra-effect should show a decrease (or increase) as the contra-level decreases (increases), for both amplitude and latency.

Another mechanism that could possibly explain the present finding of N1m suppression by contra-music may be "attention" related suppression. N1m could be altered by the status of the attention given to the stimulus; i.e., the N1m response is typically enhanced when the participant selectively pays attention to the stimulus and decreases when he/she pays attention to other stimuli such as another sound or visual stimuli [36–39]. In the present study, participants were instructed to pay attention to the tone burst to elicit N1m and to press the response button after they heard burst stimuli presented to the left ear. Therefore, in explaining the present results from the viewpoint of an attention-related phenomenon, the contra-music interfered with participants' attention to the tone burst, thus decreasing the strength of attention to the tone burst. As a result, the N1m amplitudes and latencies would become smaller and longer, respectively, than those without the contra-sound condition. The relatively smaller effects of contra-noise in the present study can be attributed to the different powers of noise and music to interfere with attention to the tone burst used, which may be related to differences in "saliency" between these two contra-stimuli [40–43]; i.e., we would expect the saliency of music stimuli (which has large fluctuations in frequency and sound level components on the time axis) to be much larger than that of continuous noise. If so, the significant effects of contra-level on N1m amplitude change observed in the MEG study (Figs 4 and 5) may reflect an increased saliency of contra-music with increasing level. It is not clear why significant effects of the contra-music level on N1m were observed for N1m amplitude but not for N1m latency. Further investigation is required to determine whether this might be a characteristic finding of the effects related to the attention and/or saliency mechanism.

The suppressive effects of the contra-music stimuli observed in the present MEG study were also confirmed by the psychophysical measurements on the effects of contra-sound on the response time to the probe tone. This may indicate that the particular auditory input presented to the contralateral ear during the signal perception in the ipsilateral ear could affect

not only the N1m responses but also the actual perceptual process to signal stimuli, depending on the characteristics of the contra-sound. In contrast, when continuous noise was applied to the contralateral ear, the average reaction time was slightly less than in the control condition (reaction time without contra-noise condition). Although it is unclear whether this trend makes sense, or why it occurs, it might be of interest because it was sometimes observed in N1m latency in the MEG study (contra-noise effects on N1m latency obtained from the right hemisphere for 40–60 dB contra-noise (Fig 4) and those from the left hemisphere for 30–50 dB contra-noise (Fig 5)).

However, it is a major limitation of the present study that the experiment was performed using only one particular type of music material (a jazz piano piece); i.e., it is unknown whether the same results would be obtained with other types of music, such as healing music. It is important to further clarify the factors related to the strength of contralateral suppression, especially from the viewpoint of the saliency of the contralateral sound.

## Supporting information

**S1 Data. File containing the data to replicate Figs 4, 5 and 7.**
(XLSX)

## Author Contributions

**Conceptualization:** Tetsuaki Kawase.

**Data curation:** Masayuki Shirakura, Tetsuaki Kawase, Akitake Kanno.

**Formal analysis:** Masayuki Shirakura, Tetsuaki Kawase.

**Funding acquisition:** Tetsuaki Kawase.

**Investigation:** Masayuki Shirakura, Tetsuaki Kawase, Akitake Kanno.

**Methodology:** Masayuki Shirakura, Tetsuaki Kawase, Akitake Kanno.

**Project administration:** Tetsuaki Kawase.

**Resources:** Nobukazu Nakasato, Ryuta Kawashima.

**Software:** Jun Ohta.

**Writing – original draft:** Masayuki Shirakura, Tetsuaki Kawase.

**Writing – review & editing:** Akitake Kanno, Nobukazu Nakasato, Ryuta Kawashima, Yukio Katori.

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
