## [Decision Letter · Decision Letter 0]

21 Sep 2021

PONE-D-21-26422Different contra-sound effects between noise and music stimuli seen in N1m and psychophysical responsesPLOS ONE

Dear Dr. Kawase,

Thank you for submitting your manuscript to PLOS ONE. After careful consideration, we feel that it has merit but does not fully meet PLOS ONE’s publication criteria as it currently stands. Therefore, we invite you to submit a revised version of the manuscript that addresses the points raised during the review process.

We look forward to receiving your revised manuscript.

Kind regards,

Paul Hinckley Delano, Ph.D.

Academic Editor

PLOS ONE

Journal Requirements:

[This study was supported by a grant from the Ministry of Education, Culture, Sports, Science and Technology-Japan (Grant-in-Aid for Exploratory Research, No. K18K195970; and a Grant-in-Aid for Scientific Research (B), No. H20H038310).]

 [1) Initials of the authors who received Grant: TK

2) Grant number: Grant-in-Aid for Exploratory Research, No. K18K195970; and a Grant-in-Aid for Scientific Research (B), No. H20H038310

3) Name of Funder: the Ministry of Education, Culture, Sports, Science and Technology-Japan

4) URL of the Funder: https://www.mext.go.jp/a_menu/shinkou/hojyo/main5_a5.htm

5) The funder had no role in study design, data collection and analysis, decision to publish, or preparation of manuscript.]

Reviewers' comments:

Reviewer's Responses to Questions

**Comments to the Author**

1. Is the manuscript technically sound, and do the data support the conclusions?

Reviewer #1: Partly

Reviewer #2: Yes

2. Has the statistical analysis been performed appropriately and rigorously? 

Reviewer #1: Yes

Reviewer #2: Yes

3. Have the authors made all data underlying the findings in their manuscript fully available?

Reviewer #1: Yes

Reviewer #2: No

4. Is the manuscript presented in an intelligible fashion and written in standard English?

Reviewer #1: Yes

Reviewer #2: Yes

5. Review Comments to the Author

Reviewer #1: This article summarizes the results of two independent experiments aimed at evaluating the "central masking phenomenon". The first experiment measured the effect of filtered noise and music on the tone-evoked magnetic fields (N1m). In the second experiment, the perceptual correlate was measured by evaluating the effect of this same contralateral acoustic stimulation (CAS) on the response time for detection of the probe sound

The investigated topic is interesting, since we do not know many aspects of central masking. The experiments are impressive to be properly executed, the manuscript is easy to read and the figures stand out for their quality. However, I believe that some aspects of the study design need to be explained in more detail in order to prepare the manuscript for publication:

1) I think the justification for the experiments should be strengthened. It is not entirely clear what specific aspect of central masking is to be studied. We know that the masking effect of music is far higher than noise (there is still controversy as to whether noise generates central masking (see Smith et al, (2000) https://doi.org/10.1121/1.428274). Therefore, the study of the CAS level seems relevant. I recommend a little more in these aspects, in order to explain the novelty of the study

2) 2) In continuation with the above, I recommend detailing how both experiments complement each other, in order to justify the relevance of both

3) I think it is important to point out why “the response time” was selected to assess central masking. Along these lines, I think it is important to detail the protocol of the second experiment, since the CAS could act as a clue to detect the ipsilateral probe stimulus (the subjects will know that after the contralateral stimulus, the ipsilateral probe tone will appear)

Additionally, I would like to point out some minor aspects that could improve the manuscript:

Methodology:

Psychoacoustic experiment:

• I recommend detailing all the stimulation parameters, it is especially relevant to know the inter-stimulus interval (was it constant or random?)

• Was the CAS synchronous with the ipsilateral stimulus? Or was it similar to the protocol used with the magnetoencephalography technique?

• Was the reaction time measured in the ipsilateral, contralateral hand or was it left to the preference of the volunteers?

• The age range is wide, was it statistically controlled in the analysis?

Results:

• In perceptual results, I recommend indicating the direction of change. According to fig. 7, on average, music appears to increase reaction time, but noise would decrease it. This dissimilar result would be interesting to be considered in the discussion

• P 21 378-382 I recommend reviewing this paragraph. I think the magnitude of the efferent effect in humans is still under study. There is some evidence that supports the idea that the contralateral efferent effect would be greater for low frequencies than high frequencies (see fig 3 on Lilaonitkul and Guinan (2009) https://doi.org/10.1007/s10162-009-0163-1 )

Reviewer #2: The manuscript presents a study that compares the masking effect of contralateral noise versus music. The authors measure this effect using MEG and reaction time. The results show that music has a significant masking effect compared to noise.

The manuscript is clear and well written. I do not see any significant methodological deficiencies. The results are well presented, and they are interesting.

Regarding the PLOS Data Policy, the authors provide the ARF amplitude and latencies changes, and response time changes. However, the raw data (MEG signals, each response time) that underlies the manuscript findings is unavailable. I will let the editor decide whether or not this is sufficient to comply with the PLOS Data Policy.

For these reasons, I suggest accepting this paper after the following issues are addressed:

* Mayor issues

1. In line 114, when you say: "The frequencies of the probe tone (250 Hz) and the contra-sound were separated as much as possible to exclude direct masking effects by the contra-sound of the probe tone.", explain why the frequency separation used in this work is, in fact, the maximum possible frequency separation. For instance, why don't you choose a 200Hz tone or a cut-off frequency of 2500 Hz?

2. Make explicit that the counter-music and counter-noise conditions are not mixed in each experimental session. Although this is clear once one reads the statement in line 125 ("In most cases..."), one may interpret, from figure 1 and the general description in the introduction, that the counter-music and counter-noise conditions might be interlaced in the same experiment.

3.Clarify if the counter-sound and counter-music sessions were counterbalanced.

4. Regarding figures 2 and 3, clarify if you are averaging AEF from counter sounds with different SPL or not.

* Minor issues

1. Specify the filter type and order for all the filters (both for the sound and MEG signals).

2. Specify the software, toolboxes, etc. used to process the MEG signals.

3. It seems to me that references to figures 2 and 3 in lines 222 and 223 are mixed. On the one hand, figure 2 is related to the counter-music condition, and figure 3 is associated with the counter-noise condition. On the other hand, line 222 says, "Superimposed magnetic signals with and without contra-music stimuli (Figs 2A and 3A) are shown ..." But, Fig 3A is related to the contra-noise stimuli. Please check this.

* Typos

1. Line 110: "Different to the study of Hari and Mäkelä 4) (1988) ..." The 4) should be [4]. Also, I'm not clear if adding the year there is consistent with the citation style.

2. Line 191: "T The influence of contralateral..." There is an extra "T."

3. Line 380: There is an extra parenthesis.

6. PLOS authors have the option to publish the peer review history of their article (what does this mean?). If published, this will include your full peer review and any attached files.

Reviewer #1: No

Reviewer #2: **Yes: **Alejandro Weinstein

---

## [Author Response · Author response to Decision Letter 0]

29 Oct 2021

Dear Editor,

We greatly appreciate the helpful suggestions and comments of the reviewers. We have revised our manuscript accordingly (revised parts are indicated in red). A point-by-point response to the editorial and reviewers' comments is provided as follows.

On the journal requirement

1) Style of the manuscript

The revised manuscript has been formatted according to PLoS One style.

2) Funding information in Acknowledgments Section 

We have removed the funding information from the acknowledgements section.

The funding information of the present study is as follows.

a) Grants received by: TK

b) Grant numbers: Grant-in-Aid for Exploratory Research, No. 18K19597; and Grant-in-Aid for Scientific Research (B), No. 20H03831.

c) Name of Funder: The Ministry of Education, Culture, Sports, Science and Technology, Japan

d) URL of the Funder: https://www.mext.go.jp/a_menu/shinkou/hojyo/main5_a5.htm

e) The funder had no role in the study design, data collection and analysis, decision to publish, or preparation of the manuscript.

3) Captions for Supporting Information files

We have added a caption for the supporting information at the end of the revised manuscript, as suggested.

“S1 Data. File containing the data to replicate Figures 4, 5 and 7.” (page 28, line 486-487).

On the Comment by Reviewer #1

1) On the justification for the experiments 

 Based on the reviewer’s suggestion, we have deepened the description of the background in the Introduction section, as follows.

“In addition, this previous study of Hari and Mäkelä [4] indicated the effect of contralateral sound on the N1m amplitude obtained from the right hemisphere for one particular sound pressure level of contra-lateral sounds, but more detailed features of these contra-sound effects such as those on N1m latencies, the effects of level of contra-sound on the magnitude of contra-sound effects (i.e., whether or not the N1 suppression effect caused by the contralateral sound is a phenomenon that depends on the presentation level of the contralateral sound), and inter-hemispheric differences have not yet been fully clarified [4].

Moreover, auditory-evoked responses can be affected by the sound presented to the contralateral ear, through peripheral mechanisms that include masking due to cross-talk and the olivocochlear (OC) efferent system that innervates the cochlear and/or the middle ear muscle system, in addition to the central masking mechanism that occurs in the brain [19-27]. Thus, it is necessary to minimize these peripheral effects during observation of the central masking effects caused by contra-sound effects, but it appears that previous studies have given little consideration to this requirement.

Against this background, the focus of the present study was to clarify in detail the features of the relatively larger contra-music effects than the contra-noise effects studied previously [4], which presumably occur mainly in the brain, while minimizing and/or assessing the possible peripheral effects caused by the presentation of contra-sound. Taking into consideration the level and/or frequency characteristics of the stimuli used in order to minimize the cross-talk effects as well as OC and MEMs effects, the effects of contra-music were compared with those of contra-noise on the latency and amplitude of N1m for various levels of contra-sound using magnetoencephalography (MEG), which can separate the activation of auditory cortices in the right and left hemispheres [8-16].” (page 5, line 73-page 6, line 97).

2) On the motivation for psychoacoustic experiments

In relation to the relation of the two experiments (MEG and psychophysical), we have added the following sentence to the Introduction.

 “Furthermore, to examine whether the phenomena observed in the N1 response are also observed psychophysically, ‐‐‐‐“ (page 6, line 98-99).

3) On the reason why “the response time” was selected to assess central masking. 

 We have added the following description to the beginning of the results of the psychophysical experiment.

“The MEG study revealed that the amplitude and latency of N1m in response to 250 Hz tone burst at 70 dB SPL were reduced and delayed, respectively, by the presentation of music material to the contralateral ear. It was speculated that these observations in the N1m response may affect the sensation of loudness and detection latency for the 70 dB tone burst in a psychophysical manner. However, as it appeared that it would be difficult to accurately evaluate slight changes in loudness sensation due to the presence or absence of consonant sound, we decided to first observe the effect on latency.” (page 21, line 370 – page 22, line 377).

4) On the protocol of the second (psychophysical) experiment

The probe sounds were presented serially 60 times (one session) at a rate of approximately one every 1500 ms (i.e., constant), and the contra-stimuli (music and noise) were presented continuously during the one measurement session of 60 times.

The response button was pressed with whichever finger was easiest for the participant, with either the right or left hand.

We have revised the description of the method of psychophysical measurement as follows.

“Subjects were instructed to press the response button as soon as they detected each of the probe sounds, which were presented serially 60 times (one session) at a rate of approximately one every 1500 ms. The response button was pressed with whichever finger the volunteer found easiest, with either the right or left hand. The timing of presentation of the 250 Hz probe tones was formatted as a wav file and controlled by a PC system. The captured reactions were also recorded using the PC system. Because the first few trials can be unstable, depending on the subject, the first five trials were discarded and the average reaction time for the subsequent 55 trials was recorded as the reaction time for each measurement condition. As in the MEG study, the probe tone was presented to the left ear and the contra-sound (2000 Hz high-pass filtered white noise or 2000 Hz high-pass filtered music stimuli) was presented continuously to the right ear during each of 60 measurements (one session) through a headphone system (MDR-CD900ST; Sony, Tokyo, Japan) via USB interfaces (Rubix 22; Roland, Hamamatsu, Japan; and Komplete Audio 6; Native Instruments, Berlin, Germany).” (page 13, line 216 – 230).

5) On the age range and statistical control in the analysis

There was no significant difference between the ages of subjects who participated in the MEG study and those who participated in the psychoacoustic test. However, it was not statistically controlled in the analysis.

6) On the trend of the reaction time to shorten under the condition of noise presentation to the contralateral ear (Fig. 7) 

 To clarify this point, we have added the following sentences to the Discussion.

“In contrast, when continuous noise was applied to the contralateral ear, the average reaction time was slightly less than in the control condition (reaction time without contra-noise condition). Although it is unclear whether this trend makes sense, or why it occurs, it might be of interest because it was sometimes observed in N1m latency in the MEG study (contra-noise effects on N1m latency obtained from the right hemisphere for 40–60 dB contra-noise (Fig. 4) and those from the left hemisphere for 30–50 dB contra-noise (Fig. 5)).” (page 27, line 472 – 478).

6) On the discussion concerning the possible OC-effects evoked by contra-sound 

In response to the suggestion of Reviewer #1, we have revised the section regarding possible OC-effects evoked by contra-sound in the Discussion, as follows.

“Contralateral sound can cause considerable stimulation of the OC efferent system [19-23]. In general, OC-mediated effects are remarkable in relatively higher frequency regions (higher than 2000–4000 Hz) in animals [29]; however, it is known that OC-mediated effects can be observed in the low frequency area in addition to the higher frequency area in humans [30-35]. In this respect, response to the 250 Hz probe tone in humans may be affected by the contra-sound via the OC-efferent system. However, considering that OC-effects are well elicited by broad-band noise including the probe-tone frequency [22, 23, 35], we speculate that it is less possible that the 2 kHz high-pass filtered contra sound used in the present study could have evoked the considerable OC-mediated effects on the probe tone at 250 Hz. In fact, no considerable effects were found in N1m responses against the 2 kHz filtered contra-noise in the present study. Based on these known characteristics of OC-mediated effects, the contribution of the OC efferent system to the suppressive effect on N1m that was observed due to contra-music in the present study is thought to be minimal.” (page 23, line 405 – page 24, line 418).

 We have also added the following two references, one of which was suggested by Reviewer #1.

34. Lilaonitkul W, Guinan JJ Jr. Human medial olivocochlear reflex: effects as functions of contralateral, ipsilateral, and bilateral elicitor bandwidths. J Assoc Res Otolaryngol. 2009; 10: 459-470.

35: Lilaonitkul W, Guinan JJ Jr. Frequency tuning of medial-olivocochlear-efferent acoustic reflexes in humans as functions of probe frequency. J Neurophysiol. 2012; 107: 1598-1611.

On the Comment by Reviewer #2

1) On the frequency separation between probe tone and contra-sound 

There is no particular reason why the probe sound frequency was 250 Hz instead of 200 Hz, but we have added the following explanation regarding separation of the probe sound frequency and the cutoff frequency.

“The frequencies of the probe tone (250 Hz) and the cut-off frequency of contra-sound were separated by 3 octaves to minimize the direct masking effects by the contra-sound on the probe tone, taking the following into consideration: the possible maximum cross-talk level of the 250 Hz component expected from the level of maximum sound pressure level of the filtered contra-stimuli, the filter slope the used in the present study, and the possible inter-aural attenuation level [29].” (page 8, line 123 – 129).

2. On the explanation of experimental procedure

In response to the reviewer’s suggestion, we have revised the legend for Fig. 1 as follows:

“Fig 1. Schema of the experimental protocol. The N1m responses to tone bursts presented to the left ear at a level of 70 dB with and without contralateral stimuli presented continuously were recorded alternately while decreasing the level of contra-stimuli from 80 to 30 dB. In most cases, the contra-music and contra-noise experiments were performed on different days, considering the mental and physical burden that they placed on the subjects.” (page 9, line 152-157).

3. On the order issue if the counter-sound and counter-music sessions were counterbalanced.

We did not counterbalance the two experimental conditions. The following description has been added to the revised manuscript.

“In all cases, the effects of contra-music on the N1m responses were measured first, and then the effects of contra-noise were measured,- - - -” (page 8, line 139 – page 9, line 141).

4. On the contra-sound level in the cases presented in Figures 2 and 3.

The contralateral sound levels of the cases presented in Figs 2 and 3 were both 50 dB. We have added the presented sound levels to the text and figure legends accordingly.

5. On the filter characteristics used in the study. 

We have added the filter characteristics and/or filter condition settings, as follows. 

“---- filtered white noise (white noise filtered with a high-pass filter (filter slope = 24 dB / octave) at 2000 Hz) and filtered music stimuli (music stimuli filtered with a high-pass filter (filter slope = 18 dB / octave) at 2000 Hz) were presented to the right ear as contralateral noise.” (page 7, line 120 – page 8, line 123).

“The MEG signal was band-pass filtered between 0.03 Hz and 400 Hz (filter slope = 12 dB / octave) sampled at 10,000 Hz.” (page 11, line 179 – 181).

“- - - - the averaged data were digitally band-pass filtered from 2.0 to 45.0 Hz (setting conditions for the high-pass filter: cut-off frequency = 2.0 Hz, filter type = hamming window, filter width = 1.29 �� 2 pi; setting conditions for the low-pass filter: cut-off frequency = 45.0 Hz, filter type = hamming window, filter width = 29 � 2 pi).” (page 11, line 186 – 190).

6. On the software, toolboxes, etc. used to process the MEG signals.

In response to the reviewer’s suggestion, we have added the following description: “All MEG signals were continuously recorded during the entire experimental duration, and later analyzed (offline) using the built-in software in the MEG system (MEG Laboratory, Ricoh).” (page 11, line 182 – 184).

7. On the description “line 222 and 223 on original version” 

 We have revised this section based on the reviewer’s suggestion, as follows:

“Figures 2 and 3 show representative examples of the effects of contralateral “music” and “noise” stimuli at 50 dB, respectively, on the N1m response obtained from the right hemisphere. Superimposed magnetic signals with and without contra-sound stimuli (Figs 2A and 3A) are shown with isofield maps and ECDs superimposed on MR images (Figs. 2B and 3B, without contra-sound stimuli; and Figs. 2C and 3C, with contra-sound stimuli).” (page 14, line 244 – 249).

8. On the “Line 110: "Different to the study of Hari and Mäkelä 4) (1988) ..."

 We have revised this section based on the reviewer’s suggestion, as follows: “Different to the study of Hari and Mäkelä [4] ----- “ (page 7, line 120).

9. On the “Line 191: "T The influence of contralateral..." 

 We have revised this section based on the reviewer’s suggestion, as follows:

“The influence of contralateral..." (page 13, line 214).

10. On the “extra parenthesis” in Line 380

We have removed the extra parenthesis.

The manuscript, in whole or in part, has not been and will not be submitted elsewhere for publication. The revised manuscript has been checked by a professional native-English-speaking scientific editor.

We hope that the revisions based on the reviewers’ comments are satisfactory, and that our manuscript is now acceptable for publication in PLoS One.

Sincerely,

Tetsuaki Kawase, M.D.

Laboratory of Rehabilitative Auditory Science

Tohoku University Graduate School of Biomedical Engineering

1-1 Seiryo-machi, Aoba-ku, Sendai, Miyagi 980-8574, Japan

TEL: +81-22-717-7303; FAX: +81-22-717-7307

E-mail: kawase@orl.med.tohoku.ac.jp

---

## [Decision Letter · Decision Letter 1]

7 Dec 2021

Different contra-sound effects between noise and music stimuli seen in N1m and psychophysical responses

PONE-D-21-26422R1

Dear Dr. Kawase,

We’re pleased to inform you that your manuscript has been judged scientifically suitable for publication and will be formally accepted for publication once it meets all outstanding technical requirements.

Kind regards,

Paul Hinckley Delano, Ph.D.

Academic Editor

PLOS ONE

Additional Editor Comments (optional):

Reviewers' comments:

Reviewer's Responses to Questions

**Comments to the Author**

1. If the authors have adequately addressed your comments raised in a previous round of review and you feel that this manuscript is now acceptable for publication, you may indicate that here to bypass the “Comments to the Author” section, enter your conflict of interest statement in the “Confidential to Editor” section, and submit your "Accept" recommendation.

Reviewer #1: All comments have been addressed

Reviewer #2: All comments have been addressed

2. Is the manuscript technically sound, and do the data support the conclusions?

Reviewer #1: Yes

Reviewer #2: Yes

3. Has the statistical analysis been performed appropriately and rigorously? 

Reviewer #1: Yes

Reviewer #2: Yes

4. Have the authors made all data underlying the findings in their manuscript fully available?

Reviewer #1: Yes

Reviewer #2: Yes

5. Is the manuscript presented in an intelligible fashion and written in standard English?

Reviewer #1: Yes

Reviewer #2: Yes

6. Review Comments to the Author

Reviewer #1: Dear authors,

I have read the new version of the manuscript and it seems to me that all the recommendations have been considered and it seems to me a more robust manuscript than the previous version. The topic is interesting, the experiments are technically well conducted and it brings new information to our field, for which I have recommended its publication,

congratulations

Reviewer #2: The authors have addressed all the comments raised in the previous review.

7. PLOS authors have the option to publish the peer review history of their article (what does this mean?). If published, this will include your full peer review and any attached files.

Reviewer #1: No

Reviewer #2: **Yes: **Alejandro Weinstein

---

## [Editor Report · Acceptance letter]

10 Dec 2021

PONE-D-21-26422R1 

Different contra-sound effects between noise and music stimuli seen in N1m and psychophysical responses 

Dear Dr. Kawase:

I'm pleased to inform you that your manuscript has been deemed suitable for publication in PLOS ONE. Congratulations! Your manuscript is now with our production department. 

Kind regards, 

on behalf of

Dr. Paul Hinckley Delano 

Academic Editor

PLOS ONE